

# Effectiveness of *Bacillus subtilis* ANT01 and *Rhizobium* sp. 11B on the control of fusarium wilt in pineapple (*Ananas comosus*)

Lourdes Adriano-Anaya, Luis Fernando Pardo-Girón, Miguel Salvador-Adriano, Miguel Salvador-Figueroa, Isidro Ovando-Medina and Benjamin Moreno-Castillo

Instituto de Biociencias, Campus IV, Universidad Autónoma de Chiapas, Tapachula, Chiapas, Mexico

## ABSTRACT

Pineapple (*Ananas comosus*) is commonly infected by *Fusarium oxysporum*, causal agent of the fusarium wilt disease. Conventionally, growers use synthetic fungicides to control the disease, which lead to environmental pollution, hazardous effects on non-target organisms and risks on human health. The aim of this work was to assess the effectiveness of *Bacillus subtilis* ANT01 and *Rhizobium* sp. 11B to control fusarium wilt on pineapple plants. Four treatments derived from a complete factorial design were tested under field conditions. Treatments composed of *B. subtilis* ANT01 and the combination *B. subtilis* ANT01–*Rhizobium* sp. 11B decreased disease severity by 94.4% and 86.1%, respectively. On the other hand, the treatment prepared with *Rhizobium* sp. 11B alone showed a reduction of 75.0%. Size of leaves and nutritional condition (SPAD units) of the biocontrol agents-treated plants showed no statistical differences. Moreover, *B. subtilis* ANT01 decreased by 46% the initial soil population of *F. oxysporum*, while *Rhizobium* sp. 11B, *B. subtilis* ANT01 plus *Rhizobium* sp. 11B and control, showed a population reduction of 12.5%, 24.2% and 23.0%, respectively. These results make evident the potential of *B. subtilis* ANT01 as biocontrol agent of the pathogen under field conditions.

# INTRODUCTION

Pineapple (*Ananas comosus*) is a bromeliad indigenous of Brazil and well-adapted to other tropical regions (*Vélez-Izquierdo et al., 2020*). The plant has a short and robust stem, thick leaves and produce up to 200 inflorescences that conforms a syncarp or infruitescence commonly known as pineapple (*Nassr & Abu Naser, 2018*). During 2021 Mexico produced 1,271,520 tons of pineapple, allowing to reach the ninth position in the world ranking of pineapple production (*Servicio de Información Agroalimentaria y Pesquera SIAP, 2022*). The state of Veracruz is the most important producer with 62.6% of the national production, followed by Oaxaca and Quintana Roo with 13.5% and 8.5%, respectively (*Secretaría de Agricultura y Desarrollo Rural, 2022a*). Pineapple orchards are affected by biotic and abiotic factors (*Moreira et al., 2016*). Some fungal pathogens that

Corresponding author
Benjamin Moreno-Castillo,
benjamin.moreno@unach.mx

infect pineapple plants are *Phytophthora nicotianae*, *Thielaviopsis paradoxa* and *Fusarium oxysporum* (*Uriza-Ávila et al., 2018*).

*F. oxysporum* is the causal agent of the fusarium wilt, a devastating disease that causes up to 80% of pineapple yield loss (*Secretaria de Agricultura y Desarrollo Rural, 2022b*). The pathogen enters the plant through wounds or natural openings in the roots and colonizes the inner vascular tissues (*Villa-Martínez et al., 2015*), where produces a polymer that accumulates and blocks water and nutrient transport from roots to aerial parts (*Hernández, Pineda & Noriega-Córdova, 2019*; *Secretaria de Agricultura y Desarrollo Rural, 2022b*),

Conventionally, fusarium wilt is controlled by the spraying of synthetic chemical fungicides, leading to environmental pollution and hazardous effects on human health and non-target organisms (*Trinidad-Cruz et al., 2017*). Alternatively, the use of antifungal microorganisms as biocontrol agents has been proposed (*Vinchira-Villarraga & Moreno-Sarmiento, 2019*).

Several works have reported the effectiveness of bacterial strains of *Bacillus* and *Rhizobium* against fusarium wilt ranging from 40 to 80% in several agricultural crops, such as banana (*Akila et al., 2011*), in tomato (*Ajilogba, Babalola & Ahmad, 2013*; *Akram et al., 2013*; *Elanchezhiyan et al., 2018*; *Patel & Saraf, 2017*; *Shanmugam & Kanoujia, 2011*), cucumber (*Cao et al., 2011*), chickpea (*Zaim, Bekkar & Belabid, 2018*; *Mehmood & Khan, 2016*), bean (*Kalantari et al., 2018*; *Tewari & Sharma, 2020*) and wheat (*Palazzini et al., 2016*). Otherwise, some strains of *Rhizobium* are free-living and can enhance plant growth, acting as biofertilizers (*Salvador-Figueroa et al., 2016*). Although cell concentrations tested in the reported literature ranged from $10^4$ to $10^{10}$ CFU/mL, there are few studies of *Bacillus* and *Rhizobium* as biocontrol agents of the fusarium wilt in pineapple orchards. Based on the former, the aim of this work was to evaluate the effectiveness of *Bacillus subtilis* ANT01 and *Rhizobium* sp. 11B strains against the fusarium wilt in pineapple under field conditions.

## MATERIALS & METHODS

### Plant material and experimental site

Pineapple shoots of about 30 cm high were collected in the municipality of Frontera Hidalgo in Chiapas state, southern Mexico (14°46′51.2″N 92°10′55.2″O). Shoots were planted at the Agroecological unit "Ayol" located in the municipality of Tapachula, Chiapas (14°49′45.3″N 92°17′48.5″O). Spacing between rows was 200 cm and 150 cm between plants, burrowed at a depth of 20 cm. Emerging weeds were manually removed and pineapple plants were irrigated when required to ensure optimum growth.

### Production of biofertilizers and biofertilization

Four liters of biol (organic liquid biofertilizer) were weekly applied per plant by drenching as reported by *Adriano et al. (2012)* plus two kilograms of bocashi added bimonthly per plant. Bocashi was prepared by homogenously mixing three layers composed each by 200 kg of coffee pulp, 200 kg of leaves and pseudostem of banana plants, 200 kg of fresh bovine manure, 24 kg of ash, 1.5 L sugarcane molasses, 2 L acid lactic bacterial broth culture and

**Table 1  Treatment design.**

| Treatment | *B. subtilis* ANT01 | *Rhizobium* sp. 11B |
|---|---|---|
| 1 | 0 | 0 |
| 2 | $10^8$ CFU/mL | 0 |
| 3 | 0 | $10^8$ CFU/mL |
| 4 | $10^8$ CFU/mL | $10^8$ CFU/mL |

1.5 L of yeast broth culture (*Saccharomyces cerevisiae*). In order to regulate temperature and moisture, the mixture was thoroughly mixed twice a day during the first seven days, then once a day during the following week and finally, every 72 h during the last 14 days of the 28-day-fermentation bioprocess.

## Treatments

Treatments were set under a factorial arrangement $2^2$ and a completely randomized design. The two factors (*B. subtilis* ANT 01 and *Rhizobium* sp. 11B strains) were tested at two levels (Table 1) and each treatment consisted of 25 plants.

## Inoculum production of *Bacillus subtilis* ANT01 and *Rhizobium* sp. 11B

Both bacterial strains *B. subtilis* ANT01 and *Rhizobium* sp. 11B were kindly provided by the Microbiological collection of the BioSciences Institute of the Autonomous University of Chiapas (UNACH). Strain ANT01 was cultured in potato dextrose broth (PDB) in 500 mL-flasks (24 g/L) during 96 h at pH $5.6 \pm 0.2$, 200 rpm and 28 °C. Strain 11B was cultured in nutrient broth (NB) during 12 h in 500 mL-flasks (8 g/L), at pH 7, 200 rpm and 28 °C. After incubation, cultures were serially-diluted and cell concentrations were estimated by the most probable number microbiological protocol (*Sutton, 2010*).

## Treatments application

Crude cultures (cells and extracellular metabolites) were added to a mixture of vermicompost leachates:water (2:10, V/V). The preparations were weekly sprayed on pineapple leaves with a hand-sprayer and until drip-point.

## Fusarium wilt severity

A scale of visible symptoms on leaves was used and consisted of five grades: grade 0 = healthy plants with no disease symptoms on leaves, grade 1 = leaves with a yellowish decoloration less than 25% of the leaf area, grade 2 = leaves with yellowish decoloration covering between 25 to 50% of the leaf area, 3 = yellowish decoloration between 50 and 75% of the leaf and grade 4 = decoloration above 75% of the leaf surface. With this information, severity was calculated with Equation (1):

$$\text{Severity} = \frac{\sum(\text{number of leaves on each grade} \times \text{grade of the scale})}{(\text{total leaves sampled}) \times (\text{highest grade of the scale})} \tag{1}$$

## Leaf size and nutritional status measured as units SPAD

Leaf size was obtained measuring the length (from base to apex) and width (in the mid part) of the basal leaves 2, 3 and 4 as suggested by *Gordillo-Delgado & Botero-Zuluaga (2020)*. In these same leaves, SPAD units were measured with a device SPAD-502 Plus[R] (Konica Minolta, Tokyo, Japan).

## Soil population of *F. oxysporum*

Quantifications of *F. oxysporum* in the treated-plants soil were estimated in composite samples collected in a ''zigzag'' sampling pattern through the experimental plots. One gram of sampled soil was suspended in nine mL of sterile Ringer's solution and serially diluted to $10^{-2}$. Then, 100 µL of the dilution were spread by triplicate onto Potato Dextrose Agar (39 g/L) added with 50 ppm Bengal rose in sterile petri dishes. After a nine-day-incubation at 28 °C, colonies morphologically similar to *F. oxysporum* were counted and verified under a microscope. The conidial and fruiting bodies morphology was also observed and compared with the reporting literature (*Kiffer & Morelet, 2000*).

## Data analysis

The influence of the analyzed factors (bacterial strains) and its statistical significance (subjected to ANOVA) during the stabilization stage of the fusarium wilt severity, the fungal soil population, leaf size and SPAD units, were assessed by the statistical procedure of a complete factorial design described by *Gutiérrez & de la Vara (2008)*. In cases of statistical significances, mean effects of factors and their interactions were graphically illustrated for an easier interpretation.

# RESULTS

## Fusarium wilt severity

Since the first application of the biocontrol agents and until the 56th day, an overall increasing trend in fusarium wilt severity was observed (Fig. 1). In pineapple plants treated with *B. subtilis* ANT01 (Treatment 2), *Rhizobium* sp. 11B (Treatment 3) and the combination of both strains (Treatment 4), the maximum increments of the severity were 1.40, 1.53 and 1.75-fold, respectively, as compared to their initial severity levels (mean severity = 0.36). In the other hand, leachates-treated plants (Treatment 1) maximum severity was 1.28-fold in comparison to its initial severity level. Furthermore, after 56 days of initial treatment an overall decrement of the disease severity curves was observed, and from 112th to 210th day a flattening pattern or stabilization stage was observed. At the end of the field experiment, average severity in treatments 1, 2, 3 and 4 represented 5.6%, 25.0%, 13.9% and 22.9%, respectively, in reference to their initial severity values.

Data analysis of the severity at the stabilization stage revealed that *B. subtilis* ANT01 effect and *B. subtilis* ANT01-*Rhizobium* sp. 11B interaction had negative values, thus, both strains decreased the fusarium wilt severity, while *Rhizobium* sp. 11B effect had a positive value and increased the disease severity (Table 2). Graphs in Fig. 2 show individual effects

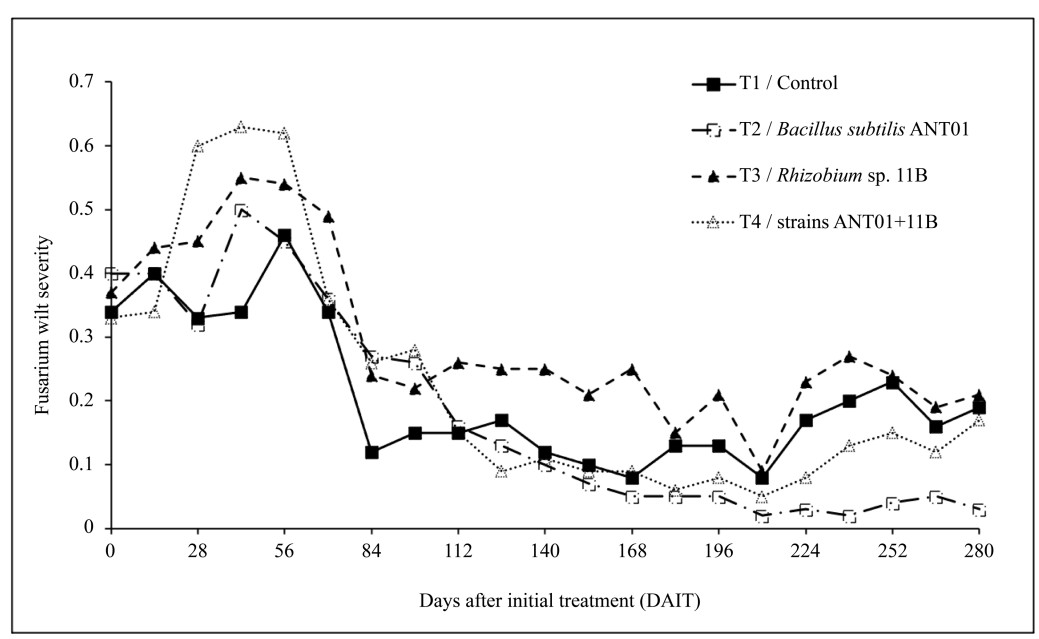

**Figure 1** Dynamics of fusarium wilt severity (*Fusarium oxysporum*) in pineapple treated plants (*Ananas comosus*).

**Table 2** Values of the principal effects of *B. subtilis* ANT01, *Rhizobium* sp. 11B and their interaction on severity of fusarium wilt in pineapple plants.

| Effect | Value |
|---|---|
| Total | 0.1325 |
| *B. subtilis* ANT01 | −0.0490 |
| *Rhizobium* sp. 11B | 0.0283 |
| *B. subtilis* ANT01 x *Rhizobium* sp. 11B | −0.0063 |

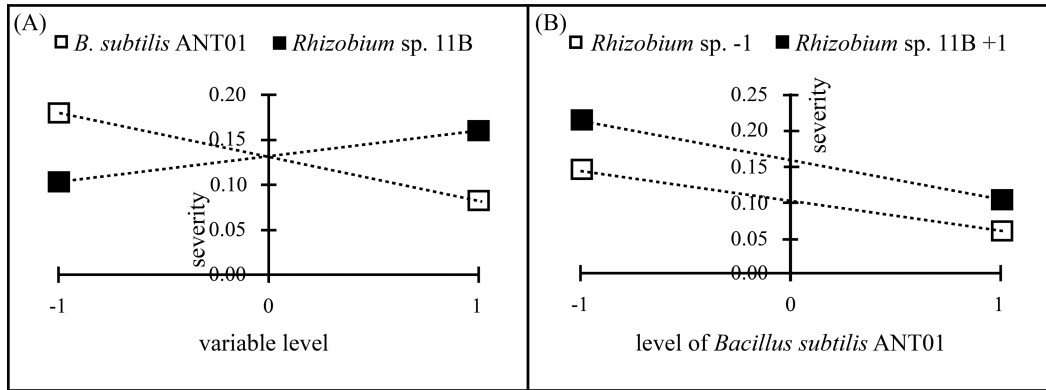

**Figure 2** Principal effects of *B subtilis*. ANT01, *Rhizobium* sp. 11B (A) and their interaction (B) on severity of fusarium wilt during the stabilization stage of the disease.

**Table 3** Comparisons of fusarium wilt severity in pineapple plants obtained experimentally and by the principal effects linearized equation of *B. subtilis* ANT01 and *Rhizobium* sp. 11B.

| Treatment | *B. subtilis* ANT01 | *Rhizobium* sp. 11B | Predicted | Experimental | Difference |
|---|---|---|---|---|---|
| 1 | −1 | −1 | 0.1469 | 0.1469 | 0.0000 |
| 2 | 1 | −1 | 0.0615 | 0.0615 | 0.0000 |
| 3 | −1 | 1 | 0.2161 | 0.2162 | −0.0001 |
| 4 | 1 | 1 | 0.1055 | 0.1054 | 0.0001 |

and the interaction of the bacterial strains used as biocontrol agents of fusarium wilt in this experiment.

The ANOVA of fusarium wilt severity revealed statistical differences in pineapple plants treated with *B. subtilis* ANT01 ($F_{1,48} = 63.9$; $P < 0.01$) and *Rhizobium* sp. 11B ($F_{1,48} = 21.2$; $P < 0.01$) but not in their interaction ($F_{1,48} = 1.1$; $P = 0.31$).

Using the effect values from Table 2 as coefficients on the linearized equation $Y = 0.1325 - 0.0490$ *B. subtilis* ANT01 + 0.0283 *Rhizobium* sp. 11B - 0.0063 *B. subtilis* ANT01-*Rhizobium* sp. 11B, predicted values of fusarium wilt severity during the stabilization stage for each treatment are given in Table 3.

The mean difference between experimental and predicted values was 0.1 thousandth, which is very low and with a determination coefficient $R^2$ of 0.6425 ($R^2 = [(SS_{Total} - SS_{Error})/(SS_{Total})]$; taken from ANOVA of severity, not shown).

## Leaf size and nutritional condition (SPAD units) of pineapple plants

Throughout the experiment, the overall average leaf length of pineapple plants was 44.94 cm (ranging from 43.35 cm in plants from treatment *B. subtilis* ANT01 to 46.16 cm in plants from treatment *Rhizobium* sp. 11B). In addition, the overall average leaf width of pineapple plants was 3.5 cm (ranging from 3.15 cm in plants from treatment *B. subtilis* ANT01 to 3.87 cm in plants from treatment *Rhizobium* sp. 11B). No significant differences were detected in the ANOVA regarding leaf length (L) and leaf width (W) of the plants treated either with *B. subtilis* ANT01 (L: $F_{1,177} = 2.74$; $P = 0.10$; W: $F_{1,177} = 1.40$; $P = 0.24$) or *Rhizobium* sp. 11B (L: $F_{1,177} = 0.79$; $P = 0.37$; W: $F_{1,177} = 0.39$; $P = 0.53$), neither for their interaction (L: $F_{1,177} = 0.11$; $P = 0.74$; W: $F_{1,177} = 1.47$; $P = 0.23$). The determination coefficients $R^2$ obtained from ANOVA were 0.020 and 0.083 for L and W, respectively.

The effects of *B. subtilis* ANT01, *Rhizobium* sp. 11B and the interaction on leaf length and width from all treatments are shown in Table 4. The highest effects were as a result of the application of *B. subtilis* ANT01, but only represented 2.0% and 7.5% of the length and width total effects, respectively. Prediction equations were: $L = 44.94 - 0.92$ *B. subtilis* ANT01 + 0.49 *Rhizobium* sp. 11B +0.18 *B. subtilis* ANT01-*Rhizobium* sp. 11B, while $W = 3.508 - 0.263$ *B. subtilis* ANT01 + 0.098 *Rhizobium* sp. 11B. The proportions of both variables resulted in 2.2% and 22.9% of maximum difference (Table 5).

The overall average of nutritional condition in pineapple leaves (measured in SPAD units from leaves 2, 3 and 4) was 49.0 (ranging from 47.8 in plants from treatment *Rhizobium* sp. 11B to 51.0 in plants from control). No statistical differences were detected in the

**Table 4** Effects of *B. subtilis* ANT01, *Rhizobium* sp. 11B, and their interaction on leaf length and width of pineapple plants.

| Effect | Length (cm) | Width (cm) |
|---|---|---|
| Total | 44.9372 | 3.5078 |
| *B. subtilis* ANT01 | −0.9161 | −0.2633 |
| *Rhizobium* sp. 11B | 0.4917 | 0.0978 |
| *B. subtilis* ANT01 x *Rhizobium* sp. 11B | 0.1806 | 0.0000 |

**Table 5** Average values of length and width of pineapple plants obtained from the prediction equations and experimentally.

| *B. subtilis* ANT01 | *Rhizobium* sp. 11B | Leaf length (cm) | | | Leaf width (cm) | | |
|---|---|---|---|---|---|---|---|
| | | Predicted | Experimental | Difference | Predicted | Experimental | Difference |
| −1 | −1 | 44.33 | 45.54 | 1.21 | 3.34 | 3.67 | 0.33 |
| 1 | −1 | 46.35 | 46.16 | −0.18 | 3.87 | 3.15 | −0.72 |
| −1 | 1 | 43.71 | 43.35 | −0.36 | 3.15 | 3.87 | 0.72 |
| 1 | 1 | 45.18 | 44.69 | −0.49 | 3.67 | 3.34 | −0.33 |

**Table 6** Effects of *B. subtilis* ANT01, *Rhizobium* sp. 11B, and their interaction on SPAD units of pineapple leaves, as predicted from the SPAD equation and experimentally.

| | Effect | Predicted | Experimental | Difference |
|---|---|---|---|---|
| Total | 49.00 | 47.01 | 47.75 | 0.74 |
| *B. subtilis* ANT01 | −0.37 | 48.56 | 48.63 | 0.07 |
| *Rhizobium* sp. 11B | −0.81 | 50.24 | 50.99 | 0.75 |
| *B. subtilis* ANT01 x *Rhizobium* sp. 11B | 0.81 | 49.37 | 48.63 | −0.74 |

ANOVA of SPAD units from strain ANT01-treated plants ($F_{1,177} = 0.75$; $P = 0.39$), strain 11B-treated plants ($F_{1,177} = 3.55$; $P = 0.06$) and in ANT01-11B interaction ($F_{1,177} = 3.54$; $P = 0.06$).

The highest effect was registered in treatment *Rhizobium* sp. 11B and represented 1.6% of the total effect, and the highest deviation derived from prediction equation (SPAD = 48.999–0.372 *B. subtilis* ANT01 - 0.810 *Rhizobium* sp. 11B + 0.809 *B. subtilis* ANT01-*Rhizobium* sp. 11B) was 1.5% (Table 6). The determination coefficient for SPAD units was 0.0424.

## Colonies of *Fusarium oxysporum* from soil cultivated with pineapple plants

Dynamics of *F. oxysporum* growing colonies are shown in Fig. 3. Twenty-eight days after the initial treatment (DAIT) with *Rhizobium* sp. 11B-*B. subtilis* ANT01 (T4), *Rhizobium* sp. 11B (T3) and control (T1) the *F. oxysporum* population decreased by 25%, 25% and 17%, respectively, while treatment with *B. subtilis* ANT01 (T2) increased by 40%. At the 56th day there was a peak population of 1.7, 1.6 and 2.8-fold in reference to initial values
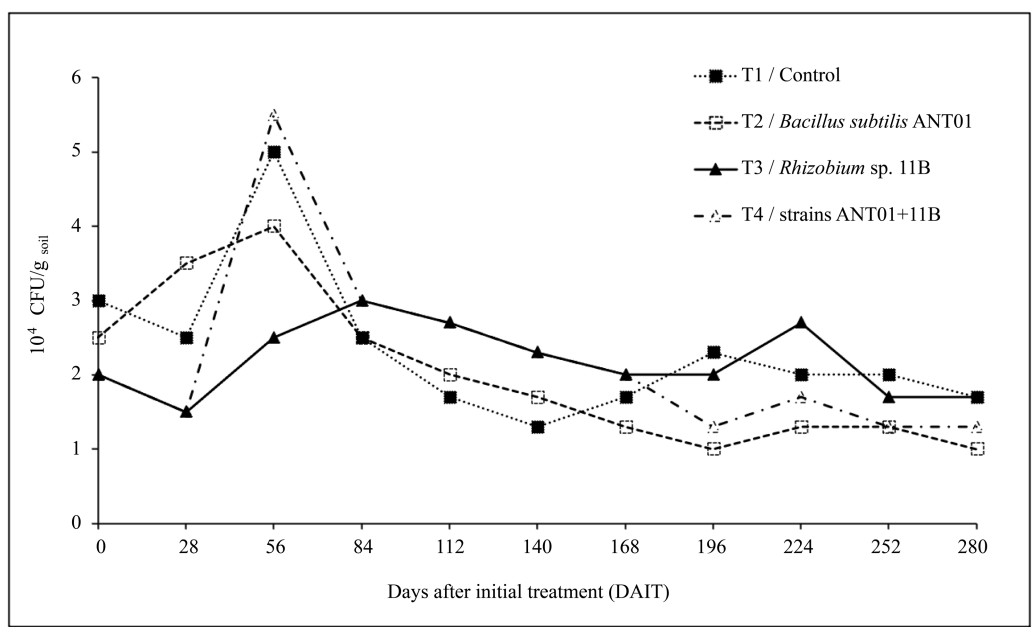

**Figure 3** Dynamics of estimated *F. oxysporum* populations in a soil grown with pineapple plants (*A. comosus*) treated with the bacterial strains ANT01 and 11B, alone or in combination.

**Table 7 Values of the principal effects of *B. subtilis* ANT01, *Rhizobium* sp. 11B and their interaction on *F. oxysporum* population in a soil cultivated with pineapple plants.**

| Effect | Value |
|---|---|
| Total | 1.7042 |
| *B. subtili* s ANT01 | −0.2458 |
| *Rhizobium* sp. 11B | 0.1542 |
| *B. subtilis ANT01 x Rhizobium* sp. 11B | 0.0375 |

of treatments T1, T2 and T4, respectively. In the case of treatment T3 the peak was at 84 DAIT, with 1.5-fold of its initial value.

In all treatments, *F. oxysporum* soil population decreased after their maximum values and after 140 DAIT was less variable. The factorial design analysis of the *F. oxysporum* population at the stabilization stage, showed that the effect of *B. subtilis* ANT01 has a negative value, which indicates that the bacteria decreased the fungal population, while *Rhizobium* sp. 11B and the interaction *B. subtilis* ANT01-*Rhizobium* sp. 11B increased the fungal soil populations due to their positive effect values (Table 7). Figure 4 shows graphically the individual and interaction effects on the fungal soil populations.

The ANOVA of *F. oxysporum* soil populations revealed statistical differences in the case of *B.subtilis* ANT01 ($F_{1,20} = 11.26$; $P < 0.05$) and *Rhizobium* sp. 11B ($F_{1,20} = 4.43$; $P < 0.05$) treatments, while treatment with both bacterial strains showed no statistical significance ($F_{1,20} = 0.26$; $P = 0.61$).

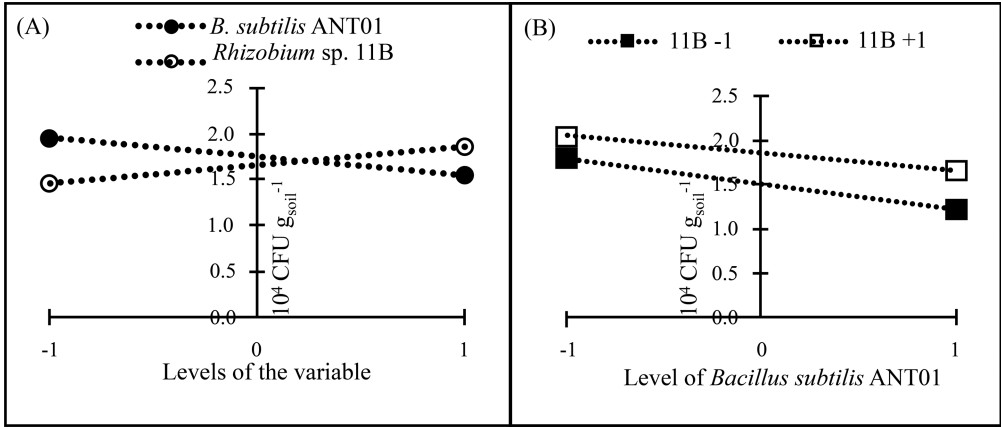

**Figure 4** Principal effects of *B subtilis* ANT01 and *Rhizobium* sp. 11B (A) and their interaction (B) on *F. oxysporum* populations from a soil cultivated with pineapple plants.

**Table 8** Effects of *B. subtilis* ANT01, *Rhizobium* sp. 11B, and their interaction on *F. oxysporum* population, as derived from the prediction equation and experimentally.

| Treatment | *B. subtilis* ANT01 | *Rhizobium* sp. 11B | Predicted | Experimental | Difference |
|---|---|---|---|---|---|
| 1 | −1 | −1 | 1.8333 | 1.8333 | 0.0000 |
| 2 | 1 | −1 | 1.2667 | 1.2667 | 0.0000 |
| 3 | −1 | 1 | 2.1045 | 1.0667 | 0.0378 |
| 4 | 1 | 1 | 1.5551 | 1.6500 | −0.0949 |

Using the effect values from Table 7 as coefficients on the linearized equation: Population of *F. oxysporum* $[10^4 \text{CFU/g}_{soil}] = 1.7042 - 0.2458$ *B. subtilis* ANT01 $+ 0.1542$ *Rhizobium* sp. 11B B $+ 0.0375$ *B. subtilis* ANT01 - *Rhizobium* sp. 11B, predicted values of the fungal soil population at the stabilization stage for each treatment are shown in Table 8.

The average difference between the experimental and predicted values was between 3.5 and 5.8% (which provided an accurate prediction) and with a determination coefficient $R^2$ of 0.4438 according to ANOVA of fungal soil populations.

## DISCUSSION

Based on the results from this work, the initial increment of fusarium wilt (increasing yellowish leaf area) observed in the pineapple plants (Fig. 1), mainly in the plants treated with the combination of *B. subtilis* ANT01 and *Rhizobium* sp. 11B, might be due to a plant immune response or priming, as mentioned by *Martínez-Medina et al. (2021)* and as demonstrated by the increasing activity of enzymes 1,3-glucanase and chitinase in tomato plants (*Solanum lycopersicum* L. var Amelia) treated with the antagonistic *Glomus mosseae* and *G. cubense* (*Pérez et al., 2015*); or the differential expression of the chalcone synthetase and phenylalanine ammonia lyase genes after the infection with the pathogen fungus *Pestalotiopsis* sp., causal agent of the gray blight in tea plants (*Camellia sinensis* L.) (*Wang et al., 2021*); or the dynamics of the salicylic acid production in strawberry plants

infected with *Podosphaera aphanis* (*Feng et al., 2020*). Alternatively, the increased severity of fusarium wilt observed in our experiment might also has been elicited as a response of *F. oxysporum* to the presence of some antifungal extracellular metabolites produced by the strains tested and/or from the naturally-living microorganisms of the vermicompost leachates. This phenomena might also be interpreted as a normal pattern of the antagonistic process, decreasing the disease severity once the biocontrol agents reached a proper density to induce the plant defenses or to act directly on the fungus. Moreover, it is not discarded that the addition of the leachates and their naturally-occurring microbiota or metabolites might have influenced the rhizosphere environment, in such a way that the treated plants produced more root exudates and promoted the presence of antifungal root-associated microorganisms (*Ren et al., 2008*; *Yuan et al., 2021*).

The levels of reduction of fusarium wilt observed in the pineapple plants of 94.4%, 75.0%, 86.1% and 77.1%, treated with *B. subtilis* ANT01 (Treatment 2), *Rhizobium* sp. 11B (Treatment 3), the combination of both strains (Treatment 4) and control (Treatment 1), respectively, are in the severity range reported for *F. oxysporum f.* sp. *lycopersici* in tomato plants treated with *B. cereus* (81.2%), *B. amyloliquefaciens* (75%), *B. pumilus* (62.5%) and *B. subtilis* (62.5%) (*Ajilogba, Babalola & Ahmad, 2013*), as well as with *Penicillium* sp. EU0013_90S (ranging from 80.6% to 95.2% of reduction) (*Hussain et al., 2016*). In reference to the pathogenic *F. oxysporum f.* sp. *spinaciae* of spinach plants treated with *F. equiseti* GF183, a range between 43.5% to 91.8% was reported (*Horinouchi, Muslim & Hyakumachi, 2010*); and in the case of *F. oxysporum f.* sp. *cubense* (Foc race 1) in banana "Prata" the reduction levels ranged from 34% to 85% (*Haddad et al., 2018*). The reduction levels reported in this work were higher than those reported for the control of *F. oxysporum f.* sp. *lycopersici* in tomato plants treated with *Bacillus* sp. ERBS10 or *B. velezensis* ERBS51 (34.9% and 50.2%, respectively) (*Devi et al., 2022*), as well as the treatment with *Penicillium* sp. EU0013_90S (46.2%) (*Alam, Sakamoto & Inubushi, 2011*) and in *F. oxysporum f.* sp. *cubense* (Foc race 1) of banana "Grand Naine" (ranging from 27.8% to 42.2%) treated with the combinations of *Pseudomonas putida* (C4r4) + *B. cereus* (Jrb1) and *Achromobacter* sp. (Gcr1) + *B. cereus* (Jrb5) (*Thangavelu & Gopi, 2015*).

Although the reduction in fusarium wilt was observed in the four treatments, data analysis showed that the treatment prepared with the strain *Rhizobium* sp. 11B promoted the symptoms disease, thus increasing the severity (as observed in Fig. 2A). Such increment may be due to the interaction plant-microbe or any metabolite that triggers the production of reactive oxygen species (ROS) provoking tissue necrosis to limit the growth of the pathogen as a defense response. Otherwise, we believe that the antagonistic activities of *B. subtilis* ANT01 and the combination *B. subtilis* ANT01 + *Rhizobium* sp. 11B (Table 2) on severity, was the result of the direct detrimental effects of the strains or their metabolites on the fungus (Figs. 1 and 2). In this regard, it has been reported that the bacteria *B. subtilis* has the ability to produce several antifungal metabolites such as iturin A, surfactin and bacilomicin D (*Gowtham et al., 2016*; *Théatre et al., 2021*). Such metabolites help on the biofilms formation, motility, and elicit cellular alterations on the fungal cell wall (*Saxena et al., 2020*). Moreover, the genus *Bacillus* is widely known for its production of an arsenal of fungal cell wall degrading enzymes (*Leelasuphakul, Sivanunsakul & Phongpaichit, 2006*)
as well as siderophores that limit the access of fungal pathogens as *F. oxysporum* to an iron source (*Goswami, Thakker & Dhandhukia, 2016*). Otherwise, since pineapple plants treated with the combination of both strains showed less severity than *Rhizobium* sp. 11B-treated plants, but higher severity levels than the *B. subtilis* ANT0-treated plants, a likely antibiotic activity might have been occurred on *Rhizobium* strain when both biocontrol agents were combined and applied on the plants. Besides, the low severity levels registered on leachates-treated plants (Treatment 1) might be as a function of some antifungal metabolites. The former is based on the determination coefficient $R^2$ of treatments below 0.9 and derived from the severity ANOVA, which allows to conclude not to be part of the experimental error.

Since leaf size (length and width) was not affected with the treatments (Tables 4 and 5), a mask-effect is likely to have occurred due to the biofertilizer effects of leachates. This was most noticeable in *Rhizobium* sp. 11B-treated plants, since the bacteria is known for its plant growth promoting traits. The lack of plant growth promoting effects observed in the pineapple plants is similar to the reported by *Devi et al. (2022)*, whose results showed a lack of promoting effects on height and total leaves of tomato plants root-inoculated with *Bacillus* sp. ERBS10 or *Bacillus velezensis* ERBS51, both with antifungal activities on *F. oxysporum f.* sp. *lycopersici*. *Haddad et al. (2018)* also reported that height and pseudostem diameter of banana plants cv. "Prata" were not increased with the inoculation of *Trichoderma harzianum*, antagonist against *F. oxysporum* f. sp. *cubense* (Foc) race 1. Contrarily, *Ajilogba, Babalola & Ahmad (2013)* reported that inoculation of tomato plants with the Foc4 antagonists *B. amyloliquefaciens*, *B. cereus*, *B. pumilus* and *B. subtilis* increased plant height and root length, while *Thangavelu & Gopi (2015)* [40] reported that banana plants cv Grand Naine inoculated with *B. flexus* (TvPr1) + *Pseudomonas putida* (Jrb2) + *B. cereus* (Jrb1), produced more bunches with more quality traits and more banana hands per bunch.

Otherwise, the lack of effect of the bacterial strains on SPAD units indicates that nitrogen fertilization in all plants was similar. Additionally, a strong argument is difficult to find regarding the best range of SPAD units recorded in this work, since to the best of our knowledge this is the first report of SPAD assessment on pineapple plants. We only may point out that SPAD units reported in this work are in the normal range reported in maize (*Novoa & Villagrán, 2002*), wheat and barley (*González, 2009*) and grape (*Castañeda et al., 2018*).

The overall time-evolution of *F. oxysporum* soil population (Fig. 3) was similar to time-evolution of fusarium wilt severity (Fig. 1), including the increasing effect during the first 56 days of the field assay. The initial increment of the pathogen soil population might be a defense response to the antifungal extracellular metabolites produced by the biocontrol agents and by the naturally-occurring microorganisms from leachates. The subsequent fungal population decrement (after 56th day) might be as a function of an increasing presence, or accumulation of antifungal metabolites up to a detrimental or suppressing concentration in the soil. Alternatively, it is likely an increment of some other fungal antagonists in the soil, due to the periodic addition of leachates as biofertilizer (*Ren et al., 2008*; *Yuan et al., 2021*).

There are few reports on the time-evolution of fungal pathogen soil populations after the treatment with biocontrol agents, and to the best of our knowledge there is no available literature in reference to pineapple plants. Nevertheless, reduction values in soil population of *F. oxysporum* reported here, were lower than the values reported by *Horinouchi, Muslim & Hyakumachi (2010)* when *F. equiseti* GF183 was applied on spinach plants to control fusariosis disease, but they indirectly determined the fungal population as a function of root weights.

Independently on the formerly mentioned, data analysis showed that *Rhizobium* sp. 11B favored the presence of fusarium wilt symptoms (severity), as observed in Fig. 4A). Likely, *Rhizobium* sp. 11B does not produce antifungal metabolites but fungal growth promoting metabolites. The negative effect of *B. subtilis* ANT01 (Table 7) may be at cause of antifungal metabolites or cell wall-degrading enzymes (*Gowtham et al., 2016*; *Théatre et al., 2021*; *Saxena et al., 2020*; *Goswami, Thakker & Dhandhukia, 2016*). In addition, the determination coefficient below 0.5 and derived from the ANOVA of fungal soil population, suggests the presence of some other fungal population detrimental factors, such as antifungal biomolecules or other microorganisms living on the vermicompost leachates.

Finally, we conclude that *B. subtilis* ANT01 is effective to reduce fusarium wilt severity and has the potential to be used as biocontrol agent of *F. oxysporum* in pineapple plants. Nevertheless, more research is required to support and clarify the whole mode of action of these bacterial biocontrol agents.

## CONCLUSIONS

We conclude that treatments composed of *B. subtilis* ANT01 and the combination *B. subtilis* ANT01–*Rhizobium* sp. 11B decreased fusarium wilt severity in pineapple plants by 94.4% and 86.1%, respectively. In addition, the treatment prepared with *Rhizobium* sp. 11B alone showed a reduction of 75.0%. Size (length and width) of leaves and their nutritional condition (SPAD units) of the biocontrol agents-treated plants showed no statistical differences. Moreover, *B. subtilis* ANT01 decreased by 46% the initial soil population of *F. oxysporum*, while *Rhizobium* sp. 11B, *B. subtilis* ANT01 plus *Rhizobium* sp. 11B and control, showed a fungal soil population reduction of 12.5%, 24.2% and 23.0%, respectively. These results make evident the potential of *B. subtilis* ANT01 as biocontrol agent of the fusarium wilt of pineapple plants under field conditions.

## ACKNOWLEDGEMENTS

Authors thank to the staff from Agroecological unit "Ayol" for all the technical support provided on the conduction of this research.

### Funding

The authors received no funding for this work.

## Competing Interests

The authors declare there are no competing interests.

## Author Contributions

- Lourdes Adriano-Anaya conceived and designed the experiments, performed the experiments, analyzed the data, prepared figures and/or tables, authored or reviewed drafts of the article, and approved the final draft.
- Luis Fernando Pardo-Girón conceived and designed the experiments, performed the experiments, analyzed the data, prepared figures and/or tables, authored or reviewed drafts of the article, and approved the final draft.
- Miguel Salvador-Adriano conceived and designed the experiments, performed the experiments, analyzed the data, prepared figures and/or tables, authored or reviewed drafts of the article, and approved the final draft.
- Miguel Salvador-Figueroa conceived and designed the experiments, performed the experiments, analyzed the data, prepared figures and/or tables, authored or reviewed drafts of the article, and approved the final draft.
- Isidro Ovando-Medina conceived and designed the experiments, performed the experiments, analyzed the data, prepared figures and/or tables, authored or reviewed drafts of the article, and approved the final draft.
- Benjamin Moreno-Castillo conceived and designed the experiments, performed the experiments, analyzed the data, prepared figures and/or tables, authored or reviewed drafts of the article, and approved the final draft.

## Data Availability

The raw data is available in the Supplementary File.

## Supplemental Information

Supplemental information for this article can be found online at http://dx.doi.org/10.7717/peerj.16871#supplemental-information.

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
