# Peer review of "Effectiveness of Bacillus subtilis ANT01 and Rhizobium sp. 11B on the control of fusarium wilt in pineapple (Ananas comosus)"

_PeerJ, doi:10.7717/peerj.16871_

## Round 0.1 · original submission · Minor Revisions

Dear Author(s),
My greetings. I appreciate your manuscript "Effectiveness of Bacillus subtilis ANT01 and Rhizobium sp. 11B on the control of fusarium wilt in pineapple (Ananas comosus)". Reviewers provided nice reviews and comments on your manuscript. Please incorporate all the suggestions.
With Best Regards
Ravindra Kumar

·

Basic reporting

The manuscript entitled ''Effectiveness of Bacillus subtilis ANT01 and Rhizobium sp. 11B on the control of fusarium wilt in pineapple (Ananas comosus)" clearly depicts the beneficial effect of bioagents for managing fusarium wilt in pineapple. Overall, the presentation is good, and the English is very much understandable. The structure of the manuscript is nicely presented with an appropriate number of figures and tables.

Experimental design

The experiments are meticulously performed with original research that meets the scope of the journal. The statistical analysis was done nicely but needs more clarification or explanation for the overall improvement of the paper. The research question is addressed properly to find out the existing knowledge gaps. Methods were performed with replication and described properly. Overall, the manuscript looks suitable for publication with a few modifications noted in the attached file

Validity of the findings

The finding is novel and has an impact on disease management. The data was robust, and statistically sound but some points need to be addressed as asked in the manuscript.

Additional comments

Few questions are required to be addressed for the improvement of the manuscript:

1. Is it possible to isolate and quantify pathogenic Fusarium oxysporum (specifically) from soil samples using Rose Bengal- PDA media? What about other contaminants as well as non-pathogenic fusaria? Please cite references if any.
2. The methodology for correlation or regression analysis should be mentioned in detail in the methodology section for better presentation of data.
3. In some cases, the value of the coefficient of determination is too low. A strong relationship can not be justified with these. Please justify.
4. In some tables, some values were mentioned. Please denote the nature or type of values (either coefficient values or something else)

·

Basic reporting

Language is clear , easy to understand, technically correct.
Literature references are cited nicely.
It is relevant topic with recent information

Experimental design

Experimental design is appropriate according to study

Validity of the findings

It is relevant topic with valid findings

Additional comments

Discussion part of this work is explained nicely. Some alignment correction is needed in tables.

---

## Round 0.2 · accepted · Accept

Dear Author(s) I found your contribution entitled as "Effectiveness of Bacillus subtilis ANT01 and Rhizobium sp. 11B on the control of fusarium wilt in pineapple (Ananas comosus)" very informative and useful for the research community. I am pleased to recommend the acceptance of the manuscript.

·

Basic reporting

The manuscript is clear and unambiguous. But some minor correction is necessary before publication.

Experimental design

Experimental design has been clearly defined. Only it is requested to write 'Soil population of Fusarium' in place of F. oxysporum. As morphologically it is very difficult to differentiate F. oxysporum from other Fusarium species.

Validity of the findings

The data is novel; only in Table 6, the predicted value is sound misleading. Please justify properly.